# Mitochondrial Heteroplasmy and PCR Amplification Bias Lead to Wrong Species Delimitation with High Confidence in the South American and Antarctic Marine Bivalve *Aequiyoldia eightsii* Species Complex

**DOI:** 10.3390/genes14040935

**Published:** 2023-04-18

**Authors:** Mariano Martínez, Lars Harms, Doris Abele, Christoph Held

**Affiliations:** 1Functional Ecology, Helmholtz Centre for Polar and Marine Research, Alfred Wegener Institute, Am Handelshafen 12, 27570 Bremerhaven, Germanychristoph.held@awi.de (C.H.); 2Oceanografía y Ecología Marina, Instituto de Ecología y Ciencias Ambientales, Facultad de Ciencias, Universidad de la República, Iguá 4225, Montevideo 11400, Uruguay; 3Helmholtz Centre for Polar and Marine Research, Alfred Wegener Institute, Am Handelshafen 12, 27570 Bremerhaven, Germany; 4Helmholtz Institute for Functional Marine Biodiversity at the University of Oldenburg (HIFMB), Ammerländer Herrstrasse 231, 26129 Oldenburg, Germany

**Keywords:** mitochondrial heteroplasmy, amplification bias, mitochondrial DNA, DNA barcoding

## Abstract

The species delimitation of the marine bivalve species complex *Aequiyoldia eightsii* in South America and Antarctica is complicated by mitochondrial heteroplasmy and amplification bias in molecular barcoding. In this study, we compare different data sources (mitochondrial cytochrome c oxidase subunit I (*COI*) sequences; nuclear and mitochondrial SNPs). Whilst all the data suggest that populations on either side of the Drake Passage belong to different species, the picture is less clear within Antarctic populations, which harbor three distinct mitochondrial lineages (p-dist ≈ 6%) that coexist in populations and in a subset of individuals with heteroplasmy. Standard barcoding procedures lead to amplification bias favoring either haplotype unpredictably and thus overestimate the species richness with high confidence. However, nuclear SNPs show no differentiation akin to the trans-Drake comparison, suggesting that the Antarctic populations represent a single species. Their distinct haplotypes likely evolved during periods of temporary allopatry, whereas recombination eroded similar differentiation patterns in the nuclear genome after secondary contact. Our study highlights the importance of using multiple data sources and careful quality control measures to avoid bias and increase the accuracy of molecular species delimitation. We recommend an active search for mitochondrial heteroplasmy and haplotype-specific primers for amplification in DNA-barcoding studies.

## 1. Introduction

The accurate delimitation and identification of species is pivotal to understanding the evolution of biodiversity and the response of communities to environmental change. DNA barcoding is a commonly used approach for assigning individuals to a known species based on standard reference DNA sequences (species identification) and for detecting still unknown species through deviations from known sequences (species delimitation). Ever since DNA barcoding was established [1], the sequencing of one or several standard DNA regions has been applied to many studies of phylogeny and phylogeography [2]. A fragment of the mitochondrial gene cytochrome c oxidase subunit I (*COI*) is one of the “gold standards” in the molecular barcoding of animals [3,4]. The utility of mitochondrial DNA (mtDNA) for species delimitation and identification is generally attributed to its high mutation rate, maternal inheritance, and lack of recombination [5].

Despite the widespread application of this convenient approach (as of February 2023, >12 million barcodes have been uploaded to the BOLD database [6]) and the invaluable progress in our understanding of global biodiversity patterns, the simplicity that prompted this technique’s success is also what limits it. Assuming a strictly uniparental form of inheritance of mtDNA through the maternal germline can overlook phenomena such as the occurrence of pseudogenes or mitochondrial heteroplasmy, leading to incorrect phylogenetic and taxonomic inferences. The risk of inadvertently using nuclear mitochondrial pseudogenes (Numts) in barcoding analyses has been considered to some degree in the barcoding literature [7,8]. However, the existence of different mitochondrial genomes in a single organism (heteroplasmy) has scarcely been suggested as a source of bias [9,10]. To the best of our knowledge, there are no concrete examples in the literature in which this peculiar feature of the mitochondrial genome has been identified as a source of misinterpretation of patterns of genetic divergence. There could be many reasons for this, but the exclusion of sequencing results characterized by the superposition of several templates in heteroplasmic organisms may be a contributing factor.

Heteroplasmy has been found in a wide range of taxa ranging from mollusks and arthropods [10,11,12] to vertebrates including fish, birds, and mammals [13,14,15]. The best-studied heteroplasmic system in metazoans is the Doubly Uniparental Inheritance (DUI) of mitochondria, an evolutionarily stable mechanism distributed across over 100 bivalve species [16,17,18]. In bivalves with DUI, females normally inherit mitochondria from their mothers (F-type), whilst males inherit mitochondria from both their mother and father (F-type and M-type). In males, the F-type is dominant in somatic tissues, whereas the M-type is prominent in the gonads but can also occur in small amounts in the somatic tissue [19]. Sex-associated mtDNAs can be extremely divergent, with up to 50% intraspecific DNA divergence [20], which is well above the divergence level typically observed among bivalve species [21].

DUI is widespread within bivalve subclasses, including among Protobranchia [22], the most basal extant bivalves with a poorly resolved phylogeny [23]. Protobranchs are of particular interest because of their high abundance in the deep oceans and important role in surface sediment carbon turnover [24]. The high-latitude marine protobranch *A. eightsii* (previously of the genus *Yoldia*) is widely distributed in the Southern Ocean, inhabiting the soft-substratum ecosystems of South America, the Antarctic Peninsula, and several Subantarctic islands. This wide distribution range on both sides of the Drake Passage has given rise to several revisions of its taxonomic status [25]. Recently, Muñoz-Ramírez et al. [26] highlighted the role of the Antarctic circumpolar current as a biogeographic barrier between the Antarctic Peninsula and South America, confirming the existence of two different species across the Drake Passage, which, according to Martínez et al. [27], have different levels of susceptibility to two major climate change stressors, namely, temperature shifts and hypoxia. González-Wevar et al. [28] suggested the presence of several cryptic species comprising two lineages along the Antarctic Peninsula (5.78% *COI* p-distance), a third linage in South America (6.5–7.5% *COI* and 1.2% ITS p-distances) with respect to the Antarctic peninsula, and two additional mitochondrial lineages on Kerguelen Island and the Falkland/Malvinas Islands. However, the scarcity of available nuclear information has hampered formal taxonomic delimitations thus far; hence, *A. eightsii* is still considered a single species.

The advances in speed and accuracy in tandem with reduced costs of next-generation sequencing (NGS) technologies have facilitated the generation of population-scale genomic surveys. Transcriptome sequencing (RNA-seq) is particularly useful for studying non-model organisms lacking reference genomes [29]. Recently, the number of studies using single-nucleotide polymorphisms (SNPs) from RNA-seq data has increased significantly, thus establishing transcriptomics as an alternative data source for population genomic studies [30,31,32]. The key analytical advantage of gaining population genomic insights from expressed sequences is the reduction in complexity afforded by limiting the sequence data to a smaller fraction devoid of repetitive and heterochromatic sequences [33].

Based on the hypothesis of heteroplasmy being a potential confounding factor in population genetics and phylogenetic analyses, this study evaluates patterns of genetic divergence between South American and Antarctic *A. eightsii* populations. Our analysis compares the results of a classic barcoding approach based on selected mitochondrial and nuclear gene fragments with SNPs derived from the deep sequencing of expressed genes, with a particular focus on how instances of heteroplasmy, if any, may affect population genetic and phylogenetic inferences.

## 2. Materials and Methods

### 2.1. Animal Collection

*A. eightsii* samples were collected in South America (Magellanic region) and Antarctic Peninsula in the austral summer season of October 2017 until February 2018. Bivalves from the Magellanic region (*n* = 100) were collected by SCUBA divers in the shallow subtidal zone near Punta Arenas (PA; Chile, 53°37′52″ S; 70°56′54″ W) on a single day in October 2017. Antarctic bivalves (*n* = 104) were collected between January and February 2018 from three sites in Potter Cove (PC), an 8 km^2^ glacial fjord on King George Island, South Shetlands (62°14′11″ S; 58°40′14″ W–62°13′32″ S; 58°38′31″ W–62°13′35″ S; 58°40′58″ W), in shallow waters between 6 and 20 m using a Van Veen grab. The maximum distance between the three collection sites in PC was 1 km, and the sites were sampled within approximately one week. An additional subset of samples (*n* = 34) collected in Hangar Cove in February 2007 (HC; near Rothera station, Adelaide Island) and Potter Cove in January 2016 and conserved at −20 °C was provided by colleagues from the British Antarctic Survey and the University of Cordoba (Argentina), respectively. These samples were exclusively used for the mitochondrial and nuclear loci analyses. In total, we obtained 238 individual organisms.

During each sampling period, bivalves were transported to the local research facility in insulated containers containing seawater and sediment from the sampling site and were dissected under a stereomicroscope after recording their individual weights, shell lengths, and widths. For each individual, three tissue types were conserved: foot and mantle tissue samples were conserved separately in RNAlater (SIGMA; St. Louis, MO, USA) and stored at −80 °C, and the rest of the soft body tissue was stored in Ethanol 95% at 4 °C until further analysis. In total, 168 bivalves were included in the analysis. All of them were utilized for the mitochondrial and nuclear marker analyses, and a subset of 70 individuals were analyzed to obtain SNPs via RNA-seq data.

### 2.2. Genomic DNA Extraction and Amplification

The molecular markers included one mitochondrial gene (cytochrome c oxidase subunit I, *COI*) amplified using Folmer primers and haplotype-specific primers (see Section 2.5 below), one nuclear protein-encoding gene (*histone H3*), and one nuclear ribosomal gene (18S ribosomal RNA, *18S*). Total genomic DNA was extracted from muscle tissue with QIAamp DNA Mini Kit (QIAGEN; Hilden, Germany) following the manufacturer’s protocol (regarding DNA purification from tissues). *COI* amplification and sequencing rwas performed for 155 individuals (*n*PA = 50, *n*PC = 82, and *n*HC = 24); *histone H3* and *18S* fragments were amplified and sequenced for 72 animals (*n*PA = 24, *n*PC = 24, and *n*HC = 24). DNA purity and concentration were assessed using a Nanodrop Spectrophotometer^®^ ND-1000 (NanoDrop Technologies; Wilmington, DE, USA). Purified genomic DNA was diluted to 10 ng/μL and used as template for Polymerase Chain Reaction (PCR) amplification. All PCRs were carried out in 25 μL reaction volumes comprising 5.0 μL 5X Colorless GoTaq^®^ Flexi Buffer (Promega Corp; Madison, WI, USA), 2 μL of MgCl2 (25 mM), 2.5 μL of dNTP mix (2 mM), 0.25 μL of Betain (5M), 0.125 of each forward and reverse primer (100 μM), 0.15 μL of GoTaq^®^ G2 Flexi DNA Polymerase (5 U/μL, PROMEGA Corp; Madison, WI, USA), and 3 μL of DNA extract (10 ng/μL). A fragment of 640 bp corresponding to *COI* was amplified using Folmer primers [3] through a PCR program consisting of the following steps: 2 min at 95 °C followed by 35 cycles at 95 °C for 20 s, 46 °C for 20 s, 72 °C for 40 s, and a final extension of 8 min at 72 °C. For the amplification of *histone H3* and *18S* fragments (257 and 752 bp, respectively), new primers were designed based on sequences of species from the same family published in [23] (*histone H3*: forward primer: 5′-GAA AAT CTA CCG GTG GCA AG-3′, reverse primer: 5′-GTG TCC TCG AAC AAA CCA AC-3′; *18S*: forward primer: 5′-AAG TAC AGA CTC TCA GTA CGG-3′, reverse primer: 5′-GAA GGC CAA CAA AAT AGA ACC-3′) and the PCR program used for both fragments consisted of the following steps: 2 min at 95 °C, 35 cycles at 95 °C for 20 s, 56 °C for 20 s, 72 °C for 40 s, and a final step of 8 min at 72 °C. After the amplification, an initial quality assessment of the PCR was conducted by placing the product on 2% agarose gel. All PCR products were sequenced for both strands by Eurofins Genomics GmbH (Ebersberg, Germany) using the same primers that were utilized during amplification. *COI*, *histone H3*, and *18S* sequences were aligned using the ClustalW algorithm of CodonCode Aligner program (version 5.1.5; CodonCode Corporation, Dedham, MA, USA), and all chromatograms were inspected visually for sequencing mistakes. For *COI*, a haplotype network was constructed using the Neighbor-Joining algorithm and implementing the Hasegawa–Kishino–Yano (HKY) model of substitution in Geneious using default parameters (version 8.1.9, Biomatters Ltd., Auckland, New Zealand) and Haplotype Viewer (Center of Integrative Bioinformatics Vienna, http://www.cibiv.at (accessed on 6 April 2022)).

### 2.3. RNA Extraction and cDNA Library Preparation

A subset of 70 individual samples out of the total included in mitochondrial and nuclear loci analysis were analyzed to obtain SNPs via RNA-seq data. A total of 70 transcriptomic libraries (one library per individual) were created in this study for animals from Punta Arenas and Potter Cove (*n*PA = 29, *n*PC = 41)

Samples of mantle tissue (5–30 mg) were homogenized in Trizol reagent (SIGMA; St. Louis, MO, USA) using a Precellys homogenizer (Precellys24, Bertin Technologies, Paris, France). Total RNA was isolated from each sample using the Direct-zolTM RNA MiniPrep Kit (ZYMO Research Corp., Irvine, CA, USA) according to the manufacturer’s instructions. The concentration and quality of the RNA were determined using a Nanodrop Spectrophotometer ND-1000 (NanoDrop Technologies, Wilmington, DE, USA) and LabChip^®^ GX Touch (PerkinElmer; Waltham, MA, USA). Libraries were prepared using the Illumina TruSeq^®^ Stranded mRNA Sample Preparation Kit starting from 1 µg of total RNA. The Poly-A-containing mRNA molecules were purified using poly-T oligo-attached magnetic beads. Subsequently, the mRNA was fragmented using divalent cations under elevated temperature and copied into the first strand of cDNA using reverse transcriptase and random primers. This process was followed by second-strand cDNA synthesis using DNA Polymerase I, RNase H, and dUTP instead of dTTP to achieve strand specificity and remove the RNA template. Following adapter ligation, products were enriched by PCR and purified to create a cDNA library. Libraries were validated and quantified using a LabChip^®^ GX Touch (PerkinElmer, Waltham, MA, USA). All the samples were pooled and cleaned using magnetic beads to remove the remaining primer content. Final cDNA concentration was measured using LabChip^®^ GX Touch (PerkinElmer, Waltham, MA, USA). The pool of samples was sequenced at the Alfred Wegener Institute on an Illumina NextSeq 500 sequencer using the NextSeq High Output Kit v2 (150 cycles) with a paired-end protocol.

### 2.4. De Novo Assembly and SNP Analysis of RNA-Seq Data

Raw reads were quality-controlled by FastQC v. 0.11.7 (Babraham Institute, Cambridge, UK). Adapter sequences were removed using bbduk.sh from the BBtools suite (version 36.38) [34] with the following parameters: ktrim = r, k = 23, mink = 11, hdist = 1, tpe, and tbo. SortMeRNA version 2.1 was used to search for rRNA sequences in the remaining sequences [35], which were removed before further processing. To filter the sequences for the common Illumina spikein PhiX, bbduk.sh was used with a kmer size of 31 and a hdist of 1. A final quality-trimming procedure was performed with bbduk.sh using Q10 as minimum quality and 36 bases as the minimum length. All obtained sequences (70 libraries) were normalized using bbnorm.sh [34] with an average depth of 100× and a minimum depth of 5×; subsequently, they were de novo assembled using the Trinity genome-independent transcriptome assembler version 2.8.4 [36] with a minimum transcript length of 300 bases and the option for strand specificity selected (--SS_lib_type RF). To remove duplicate sequences from the assembly, dedube.sh [34] was used with the following parameters: minidenty = 98, arc = t, am = t, and ac = t. The read representation and strand specificity of the assembly were assessed using the software Bowtie2 v.2.3.4.1, and completeness was evaluated using the package BUSCO v 4.1.4 (Benchmarking Universal Single-Copy Orthologs) and the orthologs of the public database “metazoa_odb10”.

For the SNP analysis, quality-filtered paired-end reads were aligned to the de novo transcriptome using bowtie2 v2.3.4.1 [37]. Alignments in SAM format (Sequence Alignment Map) were compressed and indexed with SAMtools v1.8 [38]. Genotype likelihoods were computed using mpileup from SAMtools and variant calling was performed using BCFtools. In the first filtering procedure, we excluded all variants with a Phred quality score below 30. Since our aim was to analyze SNPs, we selected only those SNPs from the variant calling that were present in at least one individual. A primary dataset of 1,308,131 SNPs in 70 individuals was filtered using VCFtools v0.1.16 [39] and by applying an iterative filtering strategy between loci and individuals with a progressive increase in cut-off values [40]. An initial filtering procedure for loci quality was performed, in which variants successfully genotyped in 50% of the individuals were kept (max-missing 0.5) and a minor allele count (MAC) of 3 was employed. Subsequently, individuals with more than 37% missing data were excluded from the analysis. A second filtering procedure for loci quality impliede a maximum missing value of 0.95, a Minor Allele Frequency (MAF) of 0.05, and a minimum number of reads (minDP) equal to 10. Additionally, variants with more than one allele were discarded (max-alleles 2). Nuclear and mitochondrial SNPs were separated by blasting the sequences containing SNPs against a mitochondrial database obtained from the UniProt Swiss-Prot database. A total of 113,515 SNPs in a final number of 54 individuals were retained. Of these, 4714 were identified as mitochondrial SNPs, while nuclear SNPs comprised all SNPs in sequences with no hits in the mitochondrial database (108,801). Mitochondrial SNPs included in the analysis (392) were those contained in a single long transcript (ca 18 kbp), representing most of the mitochondrial genome (annotated using MITOS following the approach reported by Bernt et al. [41] shown in Appendix A). To assess genetic differentiation between South American and Antarctic animals, but also between animals of different mitochondrial genotypes (mitotypes, see Section 2.5), Principal Component Analyses (PCAs), Analysis of Molecular Variance (AMOVA), and pairwise GST’ estimations were carried out separately for nuclear and mitochondrial SNPs with all the individuals that remained after the filtering process. GST’ is a corrected version of the coefficient of genetic differentiation GST, which is a quotient of heterozygosity estimates obtained from a subpopulation and the whole population. A corrected value GST’ = GST/GSTmax was computed because GST will never reach the theoretical maximum of 1 under ordinary circumstances [42]. Additional PCAs excluding the South American samples were performed to enhance differences between Antarctic mitotypes resulting from our analysis. AMOVA was conducted with respect to three strata, namely, location (South America and Antarctica), mitotype, and individuals, and significance was tested by randomly permutating (*n* = 1000) the sample matrices as described in Excoffier et al. [43]. All the analyses were performed in R v3.6 [44] using the packages vcfR [45], adegenet [46], poppr v2.8.3 [47], ade4 [48], and pegas [49].

### 2.5. Coexistence of Mitochondrial Variants in a Single Individual

Observations of raw SNP data (reads aligned to de novo transcriptome, see Section 2.4) suggested the coexistence of more than one mitochondrial haplotype (of those previously defined based on *COI* data; see Section 3.1) in a subset of individuals from Potter Cove (Antarctica). These specimens carry the haplotype h1 or h2 (never both) together with haplotype h3. To verify this finding, haplotype-specific primers were designed to selectively amplify the existing mitochondrial variants in individual samples and avoid artifacts caused by competitive differences of either mitochondrial variant in a competitive PCR using a single Folmer primer pair for both amplicons. Primer design was performed using a long transcript (ca. 18 kbp) of the de novo transcriptome representing most of the mitochondrial genome, wherein the primers were positioned in regions that were variable among haplotypes (3 bp difference). Thus, two forward primers were designed to amplify haplotype h3 (5′-AAT GTT AAT TTG TTC CAT GAG G-3′; Pb) and haplotypes h1, h2, and h4 (5′-AAT GTT AAT TTG TTC TAT GGC G-3′; Pa) and used in combination with one common reverse primer (5′-AGA AAA TAC AGC CCC CAT TC-3′) designed for both amplifications. The target amplicon (714 bp) was intentionally overlapped (by ca. 300 bp) with the *COI* sequence amplified with Folmer primers in order to track the haplotype identification based on *COI*. A subset of samples of the four haplotypes from every location (*n*h1 = 31, *n*h2 = 27, *n*h3 = 5, and *n*h4 = 16) were amplified with both primer combinations. PCR conditions for both primer combinations were the same as for *COI* with Folmer primers, with the exception of the annealing temperature (56 °C). The size of PCR products was determined on 2% agarose gel, and a representative subset of samples of the mitochondrial haplotypes successfully amplified for one or two variants (*n* = 32) was sequenced on both strands by Eurofins Genomics GmbH (Ebersberg, Germany). Finally, samples were aligned using the ClustalW algorithm of CodonCode Aligner program (version 5.1.5; CodonCode Corporation, Dedham, MA, USA). This analysis allowed us to group the animals carrying a single or two combined haplotypes in different genetic clusters (mitotypes), which was further considered for SNPs.

## 3. Results

### 3.1. Mitochondrial and Nuclear Sequence Analysis

There were no differences between the *histone H3* sequences, and only one polymorphic site was found in the *18S* fragment, which did not correspond to any geographic pattern in our sampling nor any of the clades suggested by the mitochondrial or nuclear SNP data (see Section 3.3). In most of the cases, the Sanger sequencing of the *COI* amplified using Folmer primers yielded clear sequencing results. In some cases (*n* = 13), however, ambiguous base calls that suggested the possibility of the co-existence of several *COI* alleles in the template (i.e., heteroplasmy; see 3.2 below) were observed.

The results of the *COI* sequencing with the Folmer primers show a clear differentiation between Antarctic and South American individuals (h4) but also a division between two major Antarctic clades (h1 + h2 vs. h3; Figure 1). The overall haplotype network includes 15 distinct haplotypes, 4 in South America (PA) and 9 in Antarctica, of which 7 were exclusive to PC or HC, whereas 2 were present at both Antarctic locations. The most frequent *COI* haplotypes (>2 individuals) are labeled in Figure 1 (h1, h2, h3, and h4), and their genetic distances (p-distances) are shown in Appendix A. The high p-distances between the South American haplotype h4 and the Antarctic ones (6.6–8.6%) and the distances between the rare h3 and the more frequent Antarctic h1 and h2 haplotypes (5.9–6.1%) were remarkable. These differences represent four amino acid substitutions between the Antarctic haplotypes h1 and h2 and the South American h4 and five between the Antarctic haplotype h3 and h4. No amino acid substitutions were found between the Antarctic haplotypes h1 and h2, whereas five substitutions were found between these two haplotypes and h3. No stop codons or indels were found in the *COI* sequences of any haplotype or location.

### 3.2. Coexistence of Mitochondrial Variants in a Single Organism

The co-existence of two haplotypes in a single individual was detected in several Antarctic individuals from PC (19 out of 47) and HC (1 out of 16) (h1, h2, and h3) but not in South American organisms (0 out of 16) (h4). In all but one individual that had been identified as carrying the haplotype “h3-only” with Folmer primers (see Section 3.1), the haplotypes h1 or h2 were also found once haplotype-specific primers were used. A single bivalve from HC, in which variants h1 or h2 could not be amplified, was interpreted as possessing haplotype h3 only (h3 homozygote). The reverse case, in which the Folmer primer only detected h1 or h2 but left h3 undetected, was also observed. All South American specimens (haplotype h4) and the majority of Antarctic animals analyzed qualified as homoplasmic for h1 or h2 with Folmer primers and with the newly designed haplotype specific primers (Figure 2). These findings allowed us to group the animals in six different genetic clusters based on their mitochondrial genotypes (mitotypes): animals carrying only one haplotype (h1, h2, h3, and h4) are henceforth named h1h1, h2h2, h3h3, and h4h4, respectively, and heteroplasmic animals for haplotype h1 or h2 together with the haplotype h3 are henceforth called h1h3 and h2h3, respectively. Mitotypes frequencies at each location are shown in Figure 2. This mitotype-based grouping was considered in the SNP analysis.

### 3.3. De Novo Assembly and SNP Analysis

The Illumina sequencing resulted in 454.8 million reads (average reads per library: 6.5 million), of which 388.4 million reads were used for the de novo transcriptome assembly after being filtered for quality. A total of 389,929 transcripts were assembled, with an average length of 764 bp. The BUSCO assessment of the completeness of the transcriptome showed very high quality, for which 98.7% (Complete BUSCOs: 96.6%, Fragmented BUSCOs: 2.1%) of the orthologs of the public database “metazoa_odb10” were present in the assembly.

The results of the Principal Component Analysis (PCA) for nuclear and mitochondrial SNP data are shown in Figure 3. The graphic representation of the nuclear and mitochondrial PCA includes the assignment of mitotypes to each individual plotted. Both the nuclear and mitochondrial SNP compositions clearly separate the South American animals (h4h4) from the Antarctic ones along PC1, explaining 58.6% and 88.6% of the variance, respectively. In contrast, the nuclear SNPs of the Antarctic animals were not congruent with the mitotype grouping; instead, three groups composed of animals of the four Antarctic mitotypes could be discerned. These groups, which were separated by a low percentage of variance (PC1: 5.6%, PC2: 4.8%), partially matched the sampled sites within Potter Cove, indicating close kinship due to proximity. The PCA of the mitochondrial data resulted in two clusters, one containing animals of the mitotypes h1h1 and h1h3 and the other animals corresponding to h2h2 and h2h3, with no signal reflecting the three sampling sites in Potter Cove. The removal of South American animals corresponding to h4h4 from the analysis resulted in a clearer differentiation of the Antarctic mitotypes h1h3 and h2h3 vs. h1h1 and h2h2 (PC2).

The AMOVA results confirmed the clear patterns observed in the PCAs. For mitochondrial SNPs, all three levels of variance partitioning were highly significant (*p* < 0.001), with most of the variation occurring between locations (South America and Antarctica, 89.6%) and a minor proportion occurring between mitotypes within locations (7.8%) and among individuals (2.6%). The AMOVA performed on the nuclear SNPs presented significant differences between locations (67.7%) and among individuals (31.7%) but not between mitotypes within locations. This information is summarized in Table 1.

Both the nuclear and mitochondrial pairwise genetic differentiation estimates (GST’) showed a high density of highly differentiated SNPs (GST’ = 1; i.e., present in all animals of one group and in none of the other) between animals from the two continents (Figure 4a,b), thus supporting the AMOVA, PCA, and *COI* results. An important difference between nuclear and mitochondrial GST’ estimations emerged in the form of a high density of nuclear SNPs with values close to zero in comparison to the mitochondrial data. Nuclear and mitochondrial pairwise GST’ estimates between Antarctic mitotypes are shown in Figure 4c,d. The nuclear pairwise GST’ comparisons resulted in similar density distributions, with most of the values being <0.2 and the means ranging from 0.05 to 0.09, indicating low genetic differentiation. Instead, every mitochondrial GST’ estimate comparing animals carrying the haplotype h1 and animals with h2 resulted in a high density of highly differentiated SNPs (GST’ = 1). GST’ estimations comparing h1h1 vs. h1h3 and h2h2 vs. h2h3 showed a low level of genetic differentiation (Figure 4d).

## 4. Discussion

This study calls into question the reliability of a species identification and delimitation system based on a few nuclear or mitochondrial markers such as the widely used cytochrome c oxidase subunit I (*COI*). We present a case study in a non-model marine bivalve, in which the standard barcoding approach using Folmer primers is systematically biased by the occurrence of several distinct mitochondrial genomes in the population, including heteroplasmy, which, in the absence of independent data (e.g., nuclear SNPs), would lead to an erroneous interpretation of the number of species and their distribution patterns (Figure 5). Since this systematic bias is not caused by a paucity of data, it is not averted by increasing the number of data (i.e., the number of individuals sequenced) either; instead, it might converge, as in our case and as presented in previous studies [28], on the wrong solution with high confidence. This will be especially common in non-model species whose genomic features are mostly unknown and for which inferences and indiscriminate generalization from model organisms are commonly used procedures [50].

### 4.1. Incongruence of Methods

#### 4.1.1. Comparison across the Drake Passage

Both mitochondrial and nuclear SNP analysis showed significant differences between Antarctic and South American bivalves, thus confirming the clear pattern observed in the PCAs and in the GST’ estimations. The bimodal distribution of pairwise differences in both data sources demonstrate that South American and Antarctic populations are genetically much more similar within the limits of either continental waters but highly distinct when compared between them. When viewed in conjunction with the marked differences in shell size (see Appendix A), the most likely conclusion is that populations on either side of the Drake Passage are genetically isolated and represent distinct species. Considering that the locus typicus (New South Shetland) for *A. eightsii* (Jay, 1839) is close to our sampling location (Potter Cove), it is reasonable to assume that all our Southern Ocean samples belong to *A. eightsii* (Jay, 1839) sensu strictu, whilst our South American samples (haplotype h4) belong to an as yet undescribed species that has been confused with it. However, a full taxonomical revision including the formal description of the new species of *Aequiyoldia* is beyond the scope of this paper.

#### 4.1.2. Comparison within the Southern Ocean

Whilst the extrapolation from one mitochondrial marker gene to the assumed congruent differentiation of the nuclear genome that the standard barcoding procedure implies seems justified in the comparison across the Drake Passage (Figure 4), the same is not true for emerging patterns of differentiation in the Antarctic.

In the Southern Ocean, the most notable patterns of genetic differentiation in *A. eightsii* are strongly dependent on the source of data used. Although a differentiation pattern of a magnitude similar to that of the species-level trans-Drake comparison is also present within the Southern Ocean in the mitochondrial *COI* sequence barcoding data (Figure 1; haplotypes (h1 + h2) vs. h3), the nuclear SNP data show no indication of such differentiation but show a unimodal distribution that is characteristic of a single genetic entity (Figure 4c) for all specimens regardless of their mitochondrial haplotypes (h1, h2, or h3). Furthermore, the mitochondrial *COI* barcoding sequences (derived from a 600 bp fragment of the mitochondrial *COI* gene) and the mitochondrial SNPs (derived from the 18 kb transcript representing the almost-complete mitochondrial genome) suggest a mutually incompatible differentiation pattern. The *COI* barcode sequences suggest a structure corresponding to (h1 + h2) vs. h3, whereas for the mitochondrial SNPs the most apparent division is orthogonal to the *COI* sequence data, namely, (h1h1 + h1h3) vs. (h2h2 + h2h3), possibly h3h3 by itself; thus, individuals are most strongly characterized by possessing either h1 or h2 with varying proportions of h3 admixed with either of them. This finding is not only different but incongruent and cannot easily be reconciled by assuming a different degree of taxonomic resolution in the various data sources.

#### 4.1.3. Taxonomic Consequences

The classical molecular barcoding approach based on the amplification and sequencing of a fragment of the mitochondrial *COI* gene turned out to be a good first indicator of major genetic diversity in the trans-Drake comparison. Instead, the heteroplasmic mitochondrial condition observed in some Antarctic specimens significantly interfered with its taxonomic classification when addressed by traditional molecular approaches. Considering the magnitude of the differences found by González-Wevar et al. [28] based on *COI*, it seems likely that they, too, had detected the division between h3 and (h1 + h2) and hence suggested that these two lineages were separate species, a conclusion that is not supported by our more comprehensive dataset. Our mitochondrial and nuclear SNP dataset and the coexistence of haplotype h3 with either haplotype h1 or h2 in some organisms exclude cryptic speciation between these two groups as an explanation for the observed patterns.

The widespread occurrence of mitochondrial heteroplasmy linked to Doubly Uniparental Inheritance (DUI) in bivalves and the evidence of mitochondrial heteroplasmy in a wide variety of taxa from crustaceans to vertebrates [12,15] highlight the importance of mitochondrial heteroplasmy as a general, previously underestimated shortcoming of molecular barcoding methodology. There is a consensus concerning the need for multiple loci (mitochondrial and nuclear) to perform a correct taxonomic classification, and the use of advanced sequencing techniques is intended to gain space in molecular systematics to amply address this need. However, the promise of this new sequencing era has not yet fully materialized, especially not for non-model species, demonstrating a growing taxonomic bias in favor of a small minority of genetic model species [51]. DNA barcoding using mitochondrial markers is still the dominant method with respect to non-model species and will probably remain so for a while. *A. eightsii* serves as a model case demonstrating that if heteroplasmy occurs, it can lead to false taxonomic inferences with an inflated degree of confidence. Correct taxonomic delimitation and identification is essential for the monitoring and conservation of biodiversity, especially in areas of the globe particularly threatened by climate change such as Antarctica.

### 4.2. True Process: High Mitochondrial Genetic Diversity and Heteroplasmy

Although our data strongly question the validity of the extrapolation from a fragment of *COI* to the entire (nuclear) genome, the apparent division of *COI* sequences into three almost equidistant groups, namely, (h1 + h2), h3, and h4, is likely the result of a separate evolutionary history of the three in isolation from one another. The mitotype h4 is separated by the Drake Passage and the Polar Front from its nearest relatives in the Southern Ocean, a condition that remains so until today [26]. However, divergence time estimations suggest that the haplotype h4 split off around the end of the Miocene ca. 8.5 Ma [28], thus indicating a dispersal event across the already open Drake Passage between the two continents. The differentiation of the three *Aequiyoldia* clades characterized by major haplotype groups in Antarctica likely happened in the Pliocene; however, unlike their South American relatives, all three of them occur at least in partial sympatry today. The most likely scenario for the divergent patterns found in the mitochondrial and nuclear data is that the differentiated mitochondrial lineages persisted even after the groups made secondary contact, which is a consequence of their independent, non-recombining, clonal inheritance. On the other hand, the pattern of differentiation in the nuclear genome eroded away over time as a result of the recombination of the hybridizing lineages, possibly comprising a case of speciation reversal as suggested for other Antarctic invertebrates [52]. Several examples of genetic diversification caused by temporary isolation, some of them involving cryptic speciation, were described for other Southern Ocean invertebrates such as crustaceans [53,54], polychaetes [55], and echinoderms [56], among others.

## 5. Conclusions

Population genetics, molecular barcoding, and molecular systematics critically depend on extrapolations made from small subsets to larger scales. Our results show that the confidence in the correctness of the extrapolation from a *COI* fragment to the entire genome as implied in molecular barcoding may not be justified when mitochondrial heteroplasmy is involved. In our study, an amplification bias with Folmer primers in the presence of two competing templates available for amplification in a heteroplasmic scenario resulted in the amplification of either of the two existing variants (Figure 5), thereby promoting erroneous inferences with high confidence. While such a volatile amplification bias with an unpredictable outcome may seem to require several conditions and hence may be assumed to be rare, it is possibly often overlooked because its evidence is often eliminated in a quality control step that ensures clean sequences preceding the analysis.

Countermeasures against this shortcoming include an active search for double peaks in chromatograms in the Sanger sequencing results, the use of haplotype-specific primers whenever there is evidence of sequence competition, and the use of alternative sequencing methods in combination with an active search for segregating sites.

## Figures and Tables

**Figure 1 genes-14-00935-f001:**
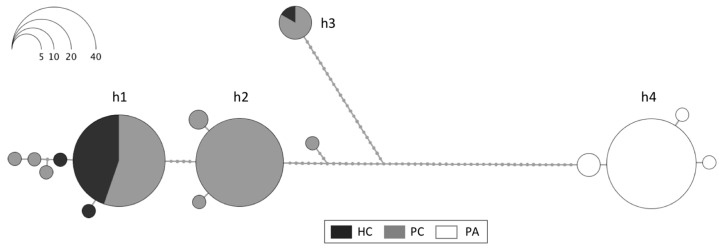
Haplotype network based on *COI* sequences of *Aequiyoldia bivalves* (*n* = 155) amplified with Folmer primers from Hangar Cove (HC), Potter Cove (PC), and Punta Arenas (PA). Each haplotype is represented by a circle proportional to its frequency (scale top left), and the most frequent haplotypes (>2 individuals) are labeled as h1, h2, h3, and h4.

**Figure 2 genes-14-00935-f002:**
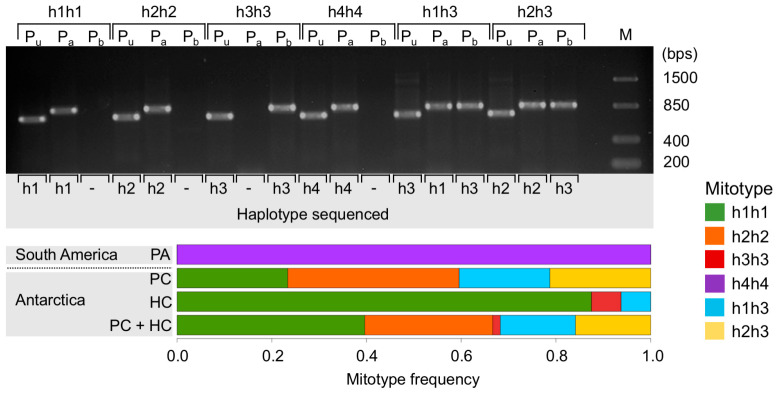
Above: Gel electrophoresis (2% agarose) of PCR-amplified products from individual samples of the six mitotypes (h1h1, h2h2, h3h3, h4h4, h1h3, and h2h3) using Folmer primers for *COI* (Pu), and the two haplotype-specific primer combinations designed in this study (Pa: h1-, h2-, and h4-specific; Pb: h3-specific). Sanger sequencing results (haplotypes) of each PCR product are shown in the bottom of the gel. Below: mitotype frequencies of bivalves from Punta Arenas (PA, *n* = 16), Potter Cove (PC, *n* = 47), Hangar Cove (HC, *n* = 16), and both Antarctic locations together (PC + HC, *n* = 63).

**Figure 3 genes-14-00935-f003:**
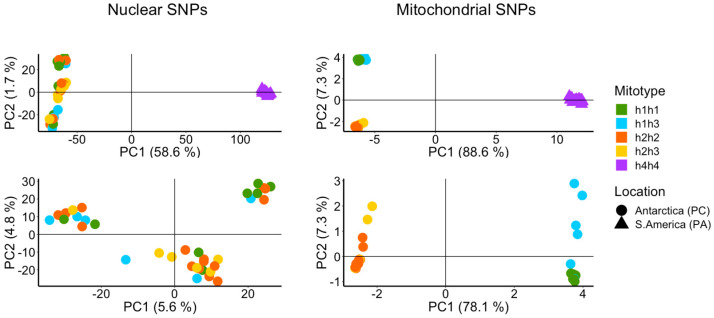
Principal component analysis based on 108,801 nuclear and 392 mitochondrial SNPs of Antarctic (circles) and South American (triangles) populations of *A. eightsii*. In panels below, only Antarctic specimens were included in the analysis. Nuclear and mitochondrial SNPs identify the samples from South America as a distinct group. Mitochondrial SNPs identify two groups among Antarctic samples characterized by haplotypes h1 or h2, respectively, either in homoplasmic condition or in heteroplasmic condition together with h3. Nuclear SNPs sort Antarctic samples into three groups that are not congruent with the mitochondrial conditions of the samples. Mitochondrial genotypes (Mitotypes) of the organisms are indicated with colors in both PCAs based on nuclear and mitochondrial SNPs.

**Figure 4 genes-14-00935-f004:**
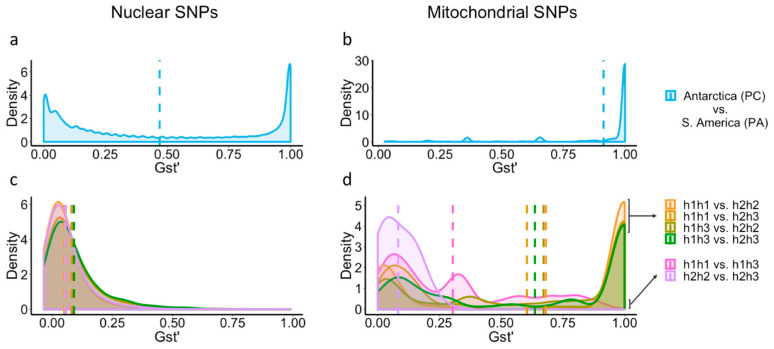
Density distributions of GST’ values estimated by pairwise comparison between South American and Antarctic populations of *A. eightsii* based on nuclear and mitochondrial SNPs (**a**,**b**) and between Antarctic mitotypes based on nuclear and mitochondrial SNPs (**c**,**d**). Dashed lines represent the mean GST’ of the correspondingly colored distribution. Note that to ease visualization, comparisons between continents do not involve comparisons within Antarctic mitotypes, and comparisons between mitotypes do not include the mitotype from South America (h4h4). A combined graphic representation would illustrate a bimodal distribution for both data sets (mitochondrial and nuclear SNPs), with most of the values close to zero for every comparison within Antarctic mitotypes and a high proportion of values close to one for every comparison between Antarctic and South American groups.

**Figure 5 genes-14-00935-f005:**
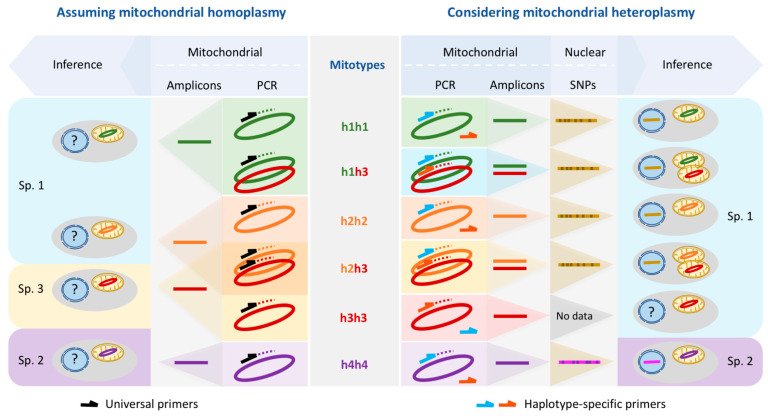
The amplification bias of Folmer primers and a standard barcoding analysis assuming strictly clonal inheritance of mitochondria leads to an overestimate of inferred species numbers (left half) in comparison to haplotype-specific primers and the inclusion of nuclear SNP data (right half). Note that amplifying h2h3 individuals using Folmer primers leads to the amplification of either h2 or h3, but never both. Although no nuclear SNP data are available for the single h3h3 individual, we include it in species 1 here on the grounds that h3 mitochondria are associated with nuclear SNP patterns that characterize species 1 (see h1h3 and h2h3).

**Table 1 genes-14-00935-t001:** Coefficients of the Analysis of Molecular Variance (AMOVA) of nuclear (Nuc) and mitochondrial (Mit) SNPs considering three strata (location, mitotype, and individual).

Source of Variation	df		Sum of Squares		Percentage of Variation		*p* Value	
	Nuc.	Mit.	Nuc.	Mit.	Nuc.	Mit.	Nuc.	Mit.
Between location	1	1	4.33	10.33	67.73	89.58	0.001	0.001
Between mitotype, within location	3	3	0.09	0.30	0.57	7.82	0.442	0.001
Within individual	49	49	0.08	0.012	31.70	2.60	0.001	0.001

df: degree of freedom.

## Data Availability

*COI* amplified with Folmer and haplotype-specific primers and *histone H3* and *18S* sequences are available in GenBank (NCBI) under the accession numbers MT176797-MT176951, MT645995-MT646032, MT647927-MT647998, and MT642948-MT643019, respectively. Raw Illumina reads were deposited in the European nucleotide Archive database (EMBL-EBI) with the accessions ERR4265443 and ERR4276392–ERR4276460 under the study accession ERP122389.

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
