# Peer review of "Mitochondrial Heteroplasmy and PCR Amplification Bias Lead to Wrong Species Delimitation with High Confidence in the South American and Antarctic Marine Bivalve Aequiyoldia eightsii Species Complex"

_genes, 2023, doi:10.3390/genes14040935_

Round 1
Reviewer 1 Report
This article was interesting to read. It's an important subject, and the findings provide further more proof that both mitochondrial and nuclear markers must be taken into account when doing species delimitation and phylogenetic analyses. In my opinion, it is well written, I would suggest only one modification: the text on topic 2.2 could be changed to "Genomic DNA extraction and amplification". Figures and tables are also good.Author Response
Responses to Reviewer 1
It's an important subject, and the findings provide further more proof that both mitochondrial and nuclear markers must be taken into account when doing species delimitation and phylogenetic analyses. In my opinion, it is well written, I would suggest only one modification: the text on topic 2.2 could be changed to "Genomic DNA extraction and amplification". Figures and tables are also good.
Answer: Suggestion accepted. The title of section 2.2 was changed to 'Genomic DNA extraction and amplification'
Reviewer 2 Report
Thank you to the authors for this contribution. I enjoyed reading this article! I found it clear, easy to read, and interesting. Here, the authors had two objectives. The first was to identify whether heteroplasmy was present in Aequiyoldia eightsii in South America and Antarctica and if so, what the implications of that would be for species delimitation methods using the barcoding marker COI. Second, the authors aimed to identify the true number of species using nuclear SNP data.
The authors identified heteroplasmy and showed that using only COI for species delimitation would result in an inflated number of species. As a remedy, they suggest that future studies take the possibility of heteroplasmy into account by checking for double peaks in Sanger sequence data, using other sequencing methods, and/or adding nuclear SNP data. They conclude that the presence of multiple sympatric haplotype groups in the Antarctic region is likely the result of secondary contact. Finally, the authors recommend a taxonomic revision characterizing the South American samples as a distinct species from A. eightsii. I have no major revisions to suggest, but minor line comments can be found below.
Introduction.
Could you add some information about how this species is expected to respond to global climate change, especially ocean warming and changing currents in the Antarctic?
Line 49. The way this is currently worded is a little hard to read, consider rewording to “…the simplicity that prompted its success is limiting” or something similar.
Line 59. Subject/verb disagreement, consider changing to, “There could be many reasons for this,”.
Line 89. Edit to, “A. eightsii is still considered a single species”
Methods.
Line 190. Indicate which lab the NextSeq was in please.
Discussion.
Line 405. Please clarify what you mean by increasing the amount of data. Do you mean more samples with the same marker? Or longer sequences from the same marker? If you increase the amount of sequencing data to include more markers, the correct solution would be achieved. Therefore, I think if you are more specific in this statement, your point will come across more strongly.
Line 482. Taxonomic consequences. At the end of this paragraph, consider adding a short sentence or two about the ultimate consequences of wrong taxonomic classification. If the Antarctic species were incorrectly assumed to be multiple species, how would this affect conservation monitoring, for example. Related to this, I was wondering if this species is at risk given climate change/warming of the Antarctic.
Conclusions.
Line 508. Similarly, in this sentence it’s not clear to me what you mean by “markers are determined with high precision”. Do you mean the whole COI gene rather than smaller portions like mini-barcode sequences?
Line 509. I think it sounds stronger to say, “may not be justified”
Line 521. Maybe also add that you can add data from nuclear SNP markers.
Author Response
Responses to Reviewer 2
Thank you to the authors for this contribution. I enjoyed reading this article! I found it clear, easy to read, and interesting. Here, the authors had two objectives. The first was to identify whether heteroplasmy was present in Aequiyoldia eightsii in South America and Antarctica and if so, what the implications of that would be for species delimitation methods using the barcoding marker COI. Second, the authors aimed to identify the true number of species using nuclear SNP data.
The authors identified heteroplasmy and showed that using only COI for species delimitation would result in an inflated number of species. As a remedy, they suggest that future studies take the possibility of heteroplasmy into account by checking for double peaks in Sanger sequence data, using other sequencing methods, and/or adding nuclear SNP data. They conclude that the presence of multiple sympatric haplotype groups in the Antarctic region is likely the result of secondary contact. Finally, the authors recommend a taxonomic revision characterizing the South American samples as a distinct species from A. eightsii. I have no major revisions to suggest, but minor line comments can be found below.
Introduction.
Could you add some information about how this species is expected to respond to global climate change, especially ocean warming and changing currents in the Antarctic?
Answer: Suggestion accepted, we inserted a comment in the introduction regarding the susceptibility to climate change stressors of the two lineages/species. (lines 89-92).
Line 49. The way this is currently worded is a little hard to read, consider rewording to “…the simplicity that prompted its success is limiting” or something similar.
Answer: Suggestion accepted. The sentence was re-written (line 55).
Line 59. Subject/verb disagreement, consider changing to, “There could be many reasons for this,”.
Answer: The sentence was re-written following the reviewer suggestion (lines 64-65).
Line 89. Edit to, “A. eightsii is still considered a single species”
Answer: Suggestion accepted (line 97)
Methods.
Line 190. Indicate which lab the NextSeq was in please.
Answer: Suggestion accepted. This information was added (lines 202-203).
Discussion.
Line 405. Please clarify what you mean by increasing the amount of data. Do you mean more samples with the same marker? Or longer sequences from the same marker? If you increase the amount of sequencing data to include more markers, the correct solution would be achieved. Therefore, I think if you are more specific in this statement, your point will come across more strongly.
Answer: We agree with reviewer ´s comment. We meant more individuals sequenced for the same marker. This is now clarified in the text (lines 433-434).
Line 482. Taxonomic consequences. At the end of this paragraph, consider adding a short sentence or two about the ultimate consequences of wrong taxonomic classification. If the Antarctic species were incorrectly assumed to be multiple species, how would this affect conservation monitoring, for example. Related to this, I was wondering if this species is at risk given climate change/warming of the Antarctic.
Answer: Suggestion accepted, the following sentence was included at the end of the section 4.1.3 of the Discusion (lines 515-517): ‘A correct taxonomic delimitation and identification is essential for the monitoring and conservation of biodiversity, especially in areas of the globe particularly threatened by climate change such as Antarctica.
Conclusions.
Line 508. Similarly, in this sentence it’s not clear to me what you mean by “markers are determined with high precision”. Do you mean the whole COI gene rather than smaller portions like mini-barcode sequences?
Answer: The reviewer's comment made us realize that this sentence is more confusing than informative. It was therefore shortened to ‘Our results show that the confidence in the correctness of the extrapolation from a COI fragment to the entire genome as implied in molecular barcoding may not be justified when mitochondrial heteroplasmy is involved.´ (line 544).
Line 509. I think it sounds stronger to say, “may not be justified”
Answer: Suggestion accepted (line 546).
Line 521. Maybe also add that you can add data from nuclear SNP markers.
Answer: This is in our opinion already covered in the text when we refer to ‘the use of alternative sequencing methods’.